# Synthesis and Biological Activity Evaluation of Coumarin-3-Carboxamide Derivatives

**DOI:** 10.3390/molecules26061653

**Published:** 2021-03-16

**Authors:** Weerachai Phutdhawong, Apiwat Chuenchid, Thongchai Taechowisan, Jitnapa Sirirak, Waya S. Phutdhawong

**Affiliations:** 1Department of Science, Faculty of Liberal Arts and Science, Kamphaeng Sean Campus, Kasetsart University, Nakhon Pathom 73140, Thailand; phutdhawong@gmail.com; 2Department of Chemistry, Faculty of Science, Silpakorn University, Nakhon Pathom 73000, Thailand; chuenchid.a@gmail.com (A.C.); jitnapasirirak@gmail.com (J.S.); 3Department of Microbiology, Faculty of Science, Silpakorn University, Nakhon Pathom 73000, Thailand; tewson84@hotmail.com

**Keywords:** coumarin3-carboxamides, coumarins, pyranocoumarins, anticancer activity, antibacterial activity

## Abstract

A series of novel coumarin-3-carboxamide derivatives were designed and synthesized to evaluate their biological activities. The compounds showed little to no activity against gram-positive and gram-negative bacteria but specifically showed potential to inhibit the growth of cancer cells. In particular, among the tested compounds, 4-fluoro and 2,5-difluoro benzamide derivatives (**14b** and **14e**, respectively) were found to be the most potent derivatives against HepG2 cancer cell lines (IC_50_ = 2.62–4.85 μM) and HeLa cancer cell lines (IC_50_ = 0.39–0.75 μM). The activities of these two compounds were comparable to that of the positive control doxorubicin; especially, 4-flurobenzamide derivative (**14b**) exhibited low cytotoxic activity against LLC-MK2 normal cell lines, with IC_50_ more than 100 μM. The molecular docking study of the synthesized compounds revealed the binding to the active site of the CK2 enzyme, indicating that the presence of the benzamide functionality is an important feature for anticancer activity.

## 1. Introduction

Coumarin is one of the potent secondary metabolites in plants [1,2] and fungi [3], and it is characterized by several pharmacological properties [4]. Like decursin **1** and decursinol **2**, these coumarins have pyranocoumarin moiety (Figure 1), having been isolated from the medicinal plant Angelica genus and shown potential for treating inflammatory diseases such as cancer, hepatic fibrosis, diabetic retinopathy, and neurological disorders [5]. The dehydrated derivative of decursinol, xanthyletin **3**, has also been shown to possess several biological properties, such as anti-tumor and antibacterial activities [6]. With a benzopyrone skeleton, coumarin is versatile and can be easily transformed into a large variety of functionalized skeletons. As a result, many coumarin derivatives have been designed, synthesized, and evaluated to address broad biological activities [7] such as antibacterial [8], antifungal [9], antioxidant [10], anti-inflammatory [11], anticancer [12], anticoagulant [13], and antiviral activities [14]. The synthetic *N*-phenyl coumarin carboxamide **4a** has been designed and shown to possess high antibacterial activity against *Helicobacter pylori* (*H. pylori)*, with the minimum inhibitory concentration (MIC) = 1 µg/mL [15], while the benzyl substitution of coumarin carboxamides **4b–d** has been shown to arrest breast cancer cell (BT474 and SKBR-3) growth by inhibiting ErbB-2 and ERK1 MAP kinase. Moreover, compounds **4b**–**d** are specific to cancer cell lines, with no cytotoxicity against normal human fibroblasts [16]. In our ongoing search for novel compounds to overcome drug resistance, the diverse biological activities of coumarins have been interesting. In the current study, we designed novel pyranocoumarin-3-carboxamide derivatives with the expectation that the carboxamide part could possess active pharmacological properties **4a**–**d** and that the pyran ring moiety could also show specific biological proteins, as in the case of xanthyletin **3**. The synthesized compounds were examined to evaluate their antibacterial activity and cytotoxicity against HepG2 and HeLa cell lines. As the coumarins were attractive casein kinase 2 (CK2) inhibitors [17], molecular docking was used to study the possible interactions of novel coumarin-3-carboxamides with the CK2 enzymes.

## 2. Results and Discussion

### 2.1. Chemistry

The preparation of pyranocoumarin-3-carboxamide was applied from the previous synthetic strategies reported by Faulgues and colleagues [18] and was described in Scheme 1. Commercially available 2,4-dihydroxy benzyldehyde **5** and 3-hydroxy-3-methyl-1-butene **6** were used as the starting materials and were subjected to Lewis acid–promoted Friedel-Crafts alkylation reaction in dioxane with BF_3_-diethylate to obtain the aldehyde **7** in 53% yield as a major product together with many unidentified products [19]. The aldehyde **7** was cyclized to form a pyrano ring using 2,3-dichloro-5,6-dicyano-l,4-benzoquinone (DDQ), and the benzopyran **8** was obtained in a very good yield. The cyclization of benzopyran **8** with the malonic anhydride **9** in pyridine and aniline at room temperature for 24 h according to a previous report [20] gave a poor yield of the pyranocoumarin-3-carboxylic acid **10,** due to the difficulty of purification. The amidation of the acid **10** with aniline using *N,N’*-dicyclohexyl carbodiimide (DCC) and 4-dimethylaminopyridine (DMAP) gave the amide **12** in 11% yield after recrystallization.

To improve the yield of pyranocoumarin-3-carboxamide **12**, the coumarin-3-carboxylic acid **13** was prepared in good yield prior to amidation with appropriate anilines using hexafluorophosphate azabenzotriazole tetramethyl Uronium (HATU) and Et_3_N to obtain amide **14a**–**g** in 43%–51% yields (Scheme 2). Then, the cyclization of **14a** by refluxing with DDQ in benzene gave pyranocoumarin-3-carboxamide **12** in 66% yield (Scheme 3). To study the effect of the substituent at C3 of coumarin ring, the carboxyl group was decarboxylated using Cu powder to give coumarin **15** in 53% yield (Scheme 4). Then, the synthesized coumarins were further evaluated for their antibacterial and anticancer activities.

### 2.2. Antibacterial Activity

Coumarin derivatives **10**, **12**, **13**, **14a**–**g**, and **15** were evaluated for their antibacterial activity against *Bacillus cereus*, *Bacillus subtilis*, *Staphylococcus aureus*, *Escherichia coli*, *Salmonella typhimurium**through* using the microbroth dilution method. Penicillin G and solvent were used as positive and negative controls, respectively, and the MIC (µg/mL) values were obtained (Table 1). The results show that only compounds **10** and **13** exhibited moderate antibacterial activities against gram-positive bacteria, while the other tested compounds displayed MIC values of more than 128 µg/mL. This may be because the carboxylic acid at the C3 position played an essential role in the antibacterial activity. Compound **15**, without the carboxyl group, showed no antibacterial activity, and the carboxamides **14a**–**g**, displayed no activity. Meanwhile, coumarin-3-carboxylic acid **13** was the most active among the tested compounds, with an MIC value of 32 μg/mL against *B. cereus*; however, it was less active than the reference drug penicillin G. Moreover, all the tested compounds showed no activity against any gram-negative bacteria.

### 2.3. Anticancer Activity

All synthesized compounds were evaluated for in vitro cytotoxic activity against two cancer cell lines (HepG2 and HeLa cell lines) and normal cell lines (LLC-MK2) through an MTT assay, and the results are presented in Table 2. Most of the tested compounds displayed potent anticancer activity. The *N*-phenyl coumarin-3-carboxamides **12** and **14a** showed significantly more potency than the parent acids **10** and **13**, respectively, against HepG2 cell lines. Moreover, compound **15**, with no substituent at C3, showed better activity than the parent acid **13**. The effect of substituents on the phenyl ring was compared with the effect of substituents on the carboxamide **14a** and it was found that the phenyl bearing fluorine atoms **14b** and **14e** possessed similar effects on the potency, while, the phenyl bearing 4-chlorine and 4-bromine atoms showed less activity against both cancer cell lines. Moreover, the phenyl bearing 4-methyl and 4-methoxyl groups displayed weak activity against the test cancer cell lines. From these results, the size and electron-donating group of the *para*-substituted benzene ring may affect anticancer properties. From these tested compounds, the amide **14e** displayed the most potent anticancer activity; however, it exhibited high cytotoxic activity against the normal cell line, with IC_50_ = 1.33 μM. Interestingly, amide **14b** displayed slightly lower activity than **14e**, but it showed low cytotoxicity against the normal cell line. This compound possessed anticancer activity comparable to those of the tested anticancer drugs doxorubicin and acridine orange.

### 2.4. Molecular Docking

Casein Kinase 2 enzyme is a key player in the pathophysiology of cancer [21,22]. Using the iGEMDOCK v2.1 software [23], molecular docking was performed to investigate binding positions and intermolecular interactions between coumarin **10**, **12**, **13**, **14a–f**, and **15** and the binding site of CK2. Coumarins **10**, **12, 13**, **14a**–**f**, and **15** were docked to the active site of CK2 co-crystallized with **G12** (PDB ID: 2QC6). Moreover, **G12** was also redocked to CK2, and its total energy and hydrogen bond length were compared with those of coumarins **10**, **12**, **13**, **14a**–**f**, and **15**. The molecular docking results show that the binding position of redocked **G12** was roughly the same as that of co-crystallized **G12** in CK2 (Figure 2a). Moreover, all synthesized compounds were bound to the active site of CK2, and the binding positions were similar to that of **G12** (Figure 2b,c), and their binding energies (−118.99 to −89.19 kcal/mol) were lower than that of **G12** (−79.10 kcal/mol) (Table 3). Figure 3 also illustrates the hydrogen bond interactions in the binding of coumarin **12**, **14a**, **14b**, **14d**, and **14e** in the cavity of CK-2 compared with that of **G12**. Coumarins **14a–f** had much lower binding energy (−118.99 to −105.53 kcal/mol) than **G12**, and their *N*-phenyl ring was also bound to similar positions of the **G12** phenolic ring (Figure 2c). Key amino acid residues, including LYS68, ASN118, ASN117, and ASP175, formed a hydrogen bond with **14a**–**f**, where ASN117 and ASN118 interacted with the hydroxy group of **14a–f**, while *N*-phenyl ring interacted with LYS68 and ASP175. The substitution group on the *N*-phenyl ring of **14a**–**f** also influenced the number of hydrogen bonds in the CK2 active site. Moreover, a comparison of the halogen substitution groups on the *N*-phenyl ring showed that the Cl and Br substitution groups on the *N*-phenyl ring formed no hydrogen bond, while the F substitution group **14b** and **14e** could form hydrogen bonds with LYS68 and ASP175 (Figure 3d,f), which may be because Cl and Br are larger than F in size. Additionally, the experimental results show that the coumarins **14b** and **14e** had good anticancer activities.

## 3. Materials and Methods

### 3.1. Chemistry

General information: Solvents and reagents were purchased from commercial suppliers TCI Chemicals (Tokyo, Japan), Sigma-Aldrich (Bangalore, India), and Fluka (Dorset, UK). Structure determination was conducted by analyzing the ^1^H, ^13^C, and ^19^F NMR spectra (Bruker 300 apparatus) and the infrared (IR) spectrum was determined using PerkinElmer Frontier Fourier-transform infrared spectrometer. Melting point was conducted using Stuart SMP2 melting point apparatus and high-resolution mass spectroscopy was analyzed by Thermo scientific, Orbitrap Q Exactive Focus.

#### 3.1.1. Synthesis of 2,4-Dihydroxy-5-(3-methylbut-2-en-1-yl)-benzaldehyde **7**

First, 2,4-dihydroxy benzaldehyde **5** (1.5 g, 10.8 mmol) in dioxane (5 mL) was added to a stirred solution of 3-hydroxy-3-methyl-1-butene **6** (1.5 mL, 14.3 mmol) and boron trifluoride diethyl etherate (BF_3_-OEt_2_, 1.5 mL) in dioxane (3 mL), and stirring was continued for 2.5 h at room temperature. Dichloromethane (50 mL) was added, and the resulting solution was extracted with water (3 × 50 mL). The combined organic layer was dried over Na_2_SO_4_ before evaporation to dryness and then purified via column chromatography (silica gel, 4:1 hexane:EtOAc) to obtain a white solid of 2,4-dihydroxy-5-(3-methylbut-2-en-1-yl) benzaldehyde **7** (0.48 g, 53% yield): m.p. 124–125 °C (lit. [19] 121–123 °C), ^1^H-NMR (300 MHz, CDCl_3_): δ 11.27 (s, 1H), 9.69 (s, 1H, OH), 7.26 (s, 1H), 6.37 (s, 1H), 6.10 (s, 1H, OH), 5.30 (tt, *J* = 4.53, 1.35 Hz, 1H), 3.30 (d, *J* = 7.2 Hz, 2H), 1.77 (s, 3H), 1.61 (s, 3H) ppm.

#### 3.1.2. Synthesis of 7-Hydroxy-2,2-dimethyl-2H-chromene-6-carbaldehyde **8**

A mixture of compound **7** (1.0 g, 4.85 mmol) and DDQ (1.2 g, 5.28 mmol) in benzene (10 mL) was refluxed for 6 h, and the precipitate was filtered off. The filtrate was evaporated to dryness to afford the crude product, which was purified via column chromatography (silica gel, 10:1 hexane:EtOAc) to obtain a white solid of 7-hydroxy-2,2-dimethyl-2H-chromene-6-carbaldehyde **8** (0.91 g, 92% yield): m.p. 82–83 °C (lit. [20] 95–96 °C), ^1^H NMR (300 MHz, CDCl_3_): δ 11.41 (br, OH), 9.68 (s, 1H), 7.16 (s, 1H), 6.35 (s, 1H), 6.30 (d, *J* = 7.86 Hz, 1H), 5.59 (d, *J* = 9.93 Hz, 1H), 1.59 (s, 3H) 1.48 (s, 3H) ppm.

#### 3.1.3. Synthesis of 8,8-Dimethyl-2-oxo-2H,8H-pyrano[3,2-g]chromene-3-carboxylic acid **10**

Compound **8** (1.0 g, 4.90 mmol) and malonic acid **9** (1.0 g, 9.60 mmol) were dissolved in pyridine (5.5 mL) containing aniline (0.5 mL) and stirred for 24 h at room temperature. Afterward, the rection mixture was poured into ice-cold 10% HCl (80 mL). The yellow precipitate was washed with cold water to remove mineral acid and then air-dried to yield a yellow solid (recrystallization by 2:1:2; EtOAc:EtOH:hexane) of 8,8-dimethyl-2-oxo-2H,8H-pyrano[3,2-g]chromene-3-carboxylic acid **10** (0.23 g, 17% yield): m.p. 187–188 °C, ^1^H NMR (300 MHz, CDCl_3_): δ 8.80 (s, 1H), 7.29 (s, 1H), 6.85 (s, 1H), 6.40 (d, *J* = 10.02 Hz, 1H), 5.81 (d, *J* = 10.02 Hz, 1H), 1.52 (s, 6H) ppm., ^13^C NMR (75 MHz, CDCl_3_): δ 164.54 (C), 163.28 (C), 160.87 (C), 156.60 (C), 150.99 (CH), 132.34 (C), 127.13 (CH), 120.17 (CH), 120.06 (CH), 112.50 (C), 110.59 (C), 104.38 (CH), 79.33 (C), 28.75 (2CH_3_) ppm. IR: 3051.36 (C-H, aromatic), 2922.46 (C-H, aliphatic), 1743.12 (C=O, acid) cm^−1^; HREI-MS (*m*/*z*) calculated for C_15_H_13_O_5_ (M)^+^ 273.0758, found 273.0757.

#### 3.1.4. Synthesis of 8,8-Dimethyl-2-oxo-*N*-phenyl-2H,8H-pyrano[3,2-g]chromene-3-carboxamide **12**

A mixture of compound **10** (0.10 g, 0.36 mmol), aniline **11** (0.040 mL, 0.43 mmol), DCC (0.10 g, 0.44 mmol), and DMAP (8 mg, 0.065 mmol) in dry CH_2_Cl_2_ (5 mL) was stirred at room temperature for 18 h. Afterward, the reaction mixture was filtered, and the filtrate was evaporated under vacuum. The residue was recrystallized using EtOH to obtain a yellow solid of 8,8-dimethyl-2-oxo-*N*-phenyl-2H,8H-pyrano[3,2-g]chromene-3-carboxamide **12 (**14 mg, 11% yield): m.p. 187–188 °C, ^1^H NMR (300 MHz, CDCl_3_): δ 10.81(s, 1H), 8.89 (s, 1H), 7.75 (d, *J* = 1.14 Hz, 2H), 7.39 (t, *J* = 6.54 Hz, 2H), 7.30 (s, 1H), 7.18 (t, *J* = 1.14 Hz, 1H), 6.80 (s, 1H), 6.40 (d, *J* = 9.90 Hz, 1H), 5.78 (d, *J* = 9.96 Hz, 1H), 1.55 (s, 6H) ppm., ^13^C NMR (75 MHz, CDCl_3_): δ 162.20 (C), 160.03 (C), 159.55 (C), 156.35 (C), 148.67 (CH), 137.30 (C), 131.82 (CH), 129.02 (2CH), 126.72 (CH), 124.57 (CH), 120.53 (2CH), 120.43 (CH), 119.58 (C), 114.71 (C), 112.73 (C), 75.72 (C), 28.64 (2CH_3_) ppm. IR: 3198.94 (N-H), 3059.35 (C-H, aromatic), 2969.10 (C-H, aliphatic), 1699.85 (C=O, amide) cm^−1^; HREI-MS (*m*/*z*) calculated for C_21_H_18_O_4_N (M)^+^ 348.1230, found 348.1230.

#### 3.1.5. Synthesis of 3-Carboxy-6-(3-methyl-2-butenyl)-7-hydroxy-coumarin **13**

Compound **7** (1.0 g, 4.85 mmol) and malonic acid **9** (1.0 g, 9.60 mmol) were dissolved in pyridine (5.5 mL) containing aniline (0.5 mL), and stirred for 24 h at room temperature. Afterward, the reaction mixture was poured into ice-cold 10% HCl (80 mL), and the yellow precipitate obtained was washed with cold water to remove mineral acid and then air-dried to yield a yellow solid of 3-carboxy-6-(3-methyl-2-butenyl)-7-hydroxy-coumarin **13** (1.10g, 88% yield): m.p. 237–238 °C (lit. [20] 218–224 °C), ^1^H NMR (300 MHz, CDCl_3_+methanol-d_4_) δ 8.79 (s. 1H), 7.40 (s, 1H), 6.85 (s, 1H), 5.32 (tt, *J* = 7.4, 1.3 Hz, 1H), 3.34 (d, *J* = 7.29 Hz, 2H), 1.79 (s, 3H), 1.70 (s, 3H) ppm., ^13^C-NMR (75 MHz, CDCl_3_+methanol-d_4_): δ 163.58 (C), 162.79 (C), 162.37 (C), 154.78 (C), 150.43 (CH), 133.49 (C), 129.32 (CH), 127.92 (C), 119.16 (CH), 110.24 (C), 107.85 (C), 100.76 (CH), 26.34 (CH_2_), 24.42 (CH_3_), 16.50 (CH_3_) ppm., IR: 3303.61 (O-H), 3049.81 (C-H, aromatic), 2911.92 (C-H, aliphatic), 1733.10 (C=O, acid), 1718.83 (C=O, lactone) cm^−1^.

#### 3.1.6. General Procedure of Coumarin-3-carboxamides Preparation (**14a**–**g**)

Triethanolamine (TEA) (0.1 mL, 1.36 mmol) was added to a solution of compound **13** (70 mg, 0.26 mmol) and HATU (0.12 g, 0.36 mmol) in dry THF (5 mL), and the mixture was stirred at room temperature for 30 min. The obtained dark clear mixture was treated with aniline derivatives (**11a**–**g**) (1.2 eq.). The resulting mixture was stirred at room temperature for 18 h. Dichloromethane (50 mL) was added, and the resulting solution was extracted with sat. NaCl (3 × 30 mL). The remaining organic layer was dried over Na_2_SO_4_ before evaporation to dryness. After evaporation of the solvent in vacuo, the crude product was purified via preparative thin-layer chromatography (silica gel, 4:1 hexane:EtOAc) to give yellow solids of coumarin-3-carboxamide (**14a**–**g**)

7-Hydroxy-6-(3-methylbut-2-en-1-yl)-2-oxo-*N*-phenyl-2H-chromene-3-carboxamide **14a** (47 mg, 51% yield): m.p. 258–259 °C, ^1^H NMR (300 MHz, CDCl_3_+pyridine-d_5_): δ 10.86 (s, NH), 8.90 (s, 1H), 7.73 (d, *J* = 7.62 Hz, 2H), 7.42 (s, 1H), 7.35 (t, *J* = 7.62 Hz, 2H), 7.12 (t, *J* = 7.38 Hz, 1H), 6.84 (s, 1H), 5.40 (tt, *J* = 7.26, 1.35 Hz, 1H), 3.43 (d, *J* = 7.26 Hz, 2H), 1.81 (s, 3H), 1.74 (s, 1H) ppm., ^13^C NMR (75 MHz, CDCl_3_+pyridine-d_5_): δ 163.38 (C), 162.81 (C), 160.56 (C), 155.49 (C), 149.36 (CH), 138.08 (CH), 134.13(C), 130.04 (CH), 128.97 (2CH), 128.68 (C), 124.35 (CH), 121.14 (CH), 120.49 (2CH), 112.94 (C), 111.33 (C), 101.68 (CH), 27.85 (CH_2_), 25.84 (CH_3_), 17.85 (CH_3_) ppm., IR: 3195.27 (O-H), 2917.31 (C-H, aliphatic), 1695.54 (C=O, amide) cm^−1^; HREI-MS (*m*/*z*) calculated for C_21_H_20_O_4_N (M)^+^ 350.1387, found 350.1386.

*N*-(4-Fluorophenyl)-7-hydroxy-6-(3-methylbut-2-en-1-yl)-2-oxo-2H-chromene-3-carboxamide **14b** (45 mg, 47% yield): m.p. 259–261 °C, ^1^H NMR (300 MHz, CDCl_3_): δ 10.56 (br, NH), 8.89 (s, 1H), 7.68 (dd, *J* = 8.97, 4.92 Hz, 2H), 7.42 (s, 1H), 7.07 (dd, *J* = 9.12, 8.79 Hz 2H), 6.83 (s, 1H), 5.34 (t, *J* = 7.23 Hz, 1H), 3.35 (d, *J* = 7.29 Hz, 2H), 1.80 (s, 3H), 1.72 (s, 1H) ppm., ^13^C NMR (75 MHz, CDCl3): δ 163.04 (C), 162.57 (C), 160.72 (C), 159.64 (d, *J* = 242.5 Hz, C), 155.46 (C), 149.71 (CH), 134.43 (C), 133.84 (d, *J* = 3.0 Hz, C), 130.23 (CH), 128.80 (C), 120.91 (CH), 115.72 (d, *J* = 22.5 Hz, 2CH), 112.86 (C), 112.31 (d, *J* = 7.5 Hz, 2CH), 111.68 (C), 101.70 (CH), 27.74 (CH_2_), 25.87 (CH_3_), 17.82 (CH_3_) ppm., ^19^F NMR (282 MHz, CDCl_3_, std. TFA): −118.08 (s, 1F) ppm., IR: 3208.66 (O-H), 3155.21 (N-H), 3048.54 (C-H, aromatic), 2913.86 (C-H, aliphatic), 1698.12 (C=O, amide) cm^−1^; HREI-MS (*m*/*z*) calculated for C_21_H_19_O_4_NF (M)^+^ 368.1293, found 368.1293.

*N*-(4-Chlorophenyl)-7-hydroxy-6-(3-methylbut-2-en-1-yl)-2-oxo-2H-chromene-3-carboxamide **14c** (44 mg, 44% yield): m.p. 282–283 °C, ^1^H NMR (300 MHz, CDCl_3_+pyridine-d_5_): δ 10.92 (s, NH), 8.89 (s, 1H), 7.69 (d, *J* = 8.88 Hz, 2H), 7.37 (s, 1H), 7.30 (dd, *J* = 8.89, 2.01 Hz, 2H), 6.83 (s, 1H), 5.41 (t, *J* = 7.23 Hz, 1H), 3.43 (d, *J* = 7.23 Hz, 2H), 1.81 (s, 3H), 1.74 (s, 1H) ppm., ^13^C NMR (75 MHz, CDCl_3_+pyridine-d_5_): δ 163.65 (C), 162.81 (C), 160.64 (C), 155.57 (C), 149.50 (CH), 136.76 (C), 134.06 (C), 130.09 (CH), 129.15 (C), 128.95 (2CH), 128.83 (C), 121.64 (2CH), 121.17 (CH), 112.56 (C), 111.27 (C), 101.68 (CH), 27.85 (CH_2_), 25.83 (CH_3_), 17.84 (CH_3_) ppm. IR: 3196.03 (O-H), 3122.34 (N-H), 3073.18 (C-H, aromatic), 2911.85 (C-H, aliphatic), 1698.47 (C=O, amide) cm^−1^; HREI-MS (*m*/*z*) calculated for C_21_H_19_O_4_N^35^Cl (M)^+^ 384.0997, found 384.0996.

*N*-(4-Bromophenyl)-7-hydroxy-6-(3-methylbut-2-en-1-yl)-2-oxo-2H-chromene-3-carboxamide **14d** (52 mg, 47% yield)): m.p. 276–277 °C, ^1^H NMR (300 MHz, CDCl_3_+pyridine-d_5_): δ 10.92 (s, NH), 8.89 (s, 1H), 7.69 (d, *J* = 8.88 Hz, 2H), 7.44 (d, *J* = 9.66 Hz, 2H), 7.39 (s, 1H), 6.83 (s, 1H), 5.40 (t, *J* = 7.14 Hz, 1H), 3.43 (d, *J* = 7.26 Hz, 2H), 1.81 (s, 3H), 1.73 (s, 1H) ppm., ^13^C NMR (75 MHz, CDCl_3_+pyridine-d_5_): δ 163.67 (C), 162.80 (C), 160.65 (C), 155.57 (C), 149.51 (CH), 137.25 (C), 134.04 (C), 131.89 (2CH), 130.09 (CH), 128.83 (C), 121.96 (2CH), 121.16 (CH), 116.80 (C), 112.52 (C), 111.25 (C), 101.67 (CH), 27.85 (CH_2_), 25.83 (CH_3_), 17.84 (CH_3_) ppm. IR: 3192.94 (O-H), 3070.45 (C-H, aromatic), 2915.14 (C-H, aliphatic), 1697.23 (C=O, amide) cm^−1^; HREI-MS (*m*/*z*) calculated for C_21_H_19_O_4_N^79^Br (M)^+^ 428.0492, found 428.0492.

*N*-(2,5-Difluorophenyl)-7-hydroxy-6-(3-methylbut-2-en-1-yl)-2-oxo-2H-chromene-3-carboxamide **14e** (46 mg, 46% yield): m.p. 248–250 °C, ^1^H NMR (300 MHz, CDCl_3_+pyridine-d_5_): δ 12.27 (s, NH), 8.90 (s, 1H), 8.40 (ddd, *J* = 10.46, 6.66, 3.15 Hz, 1H), 7.43 (s, 1H), 7.04 (ddd, *J* = 9.57, 9.18, 4.89 Hz, 1H), 6.86 (s, 1H), 6.65–6.75 (m,1H), 5.41 (t, *J* = 7.35 Hz, 1H), 3.43 (d, *J* = 7.23 Hz, 2H), 1.81 (s, 3H), 1.74 (s, 1H) ppm., ^13^C NMR (75 MHz, CDCl_3_+pyridine-d_5_): δ 163.90 (C), 162.63 (C), 160.90 (C), 158.57, (d, *J* = 238.5 Hz, C), 158.55, (d, *J* = 239.3 Hz, C), 155.75 (C), 149.73 (C), 134.03 (C), 130.18 (CH), 128.88 (C), 127.65 (d, *J* = 11.3 Hz, CH), 121.19 (CH), 115.20 (dd, *J* = 27.4, 9 Hz, CH), 112.28 (C), 111.18 (C), 110.03 (dd, *J* = 24.0, 7.5 Hz, CH), 109.09 (d, *J* = 30 Hz, CH), 101.74 (CH), 27.83 (CH_2_), 25.82 (CH_3_), 17.83 (CH_3_) ppm. ^19^F NMR (282 MHz, CDCl_3_+pyridine-d_5_, std. TFA): −117.72 (d, *J* = 14.10 Hz, 1F), -136.03 (d, *J* = 14.10 Hz, 1F) ppm., IR: 3252.42 (O-H), 3130.54 (N-H), 2073.1 (C-H, aromatic), 2915.45 (C-H, aliphatic), 1701.64 (C=O, amide) cm^−1^; HREI-MS (*m*/*z*) calculated for C_21_H_17_O_4_NF_2_ (M+Na)^+^ 408.1018, found 408.1013.

7-Hydroxy-6-(3-methylbut-2-en-1-yl)-2-oxo-N-(p-tolyl)-2H-chromene-3-carboxamide **14f** (41 mg, 43% yield): m.p. 276–277 °C, ^1^H NMR (300 MHz, CDCl_3_+pyridine-d_5_): δ 10.80 (s, NH), 8.90 (s, 1H), 7.65 (d, *J* = 7.44 Hz, 2H), 7.41 (s, 1H), 7.15 (d, *J* = 8.31 Hz, 2H), 6.85 (s, 1H), 5.41 (tt, *J* = 7.32, 1.35 Hz, 1H), 3.43 (d, *J* = 7.20 Hz, 2H), 2.31 (s, 3H), 1.81 (s, 3H), 1.74 (s, 1H) ppm., ^13^C NMR (75 MHz, CDCl_3_+pyridine-d_5_): δ 163.33 (C), 162.79 (C), 160.40 (C), 155.45 (C), 149.19 (CH), 135.56 (C), 134.01 (C), 133.91 (C), 130.00 (CH), 129.47 (2CH), 121.22 (CH), 120.45 (2CH), 113.03 (C), 111.31 (C), 101.66 (CH), 27.86 (CH_2_), 25.83 (CH_3_), 20.91 (CH_3_), 17.84 (CH_3_) ppm. IR: 3187.07 (O-H), 3130.54 (N-H), 3073.13 (C-H, aromatic), 2913.70 (C-H, aliphatic), 1698.29 (C=O, amide) cm^−1^; HREI-MS (*m*/*z*) calculated for C_22_H_22_O_4_N (M)^+^ 364.1543, found 364.1540.

7-Hydroxy-*N*-(4-methoxyphenyl)-6-(3-methylbut-2-en-1-yl)-2-oxo-2H-chromene-3-carboxamide **14g** (43 mg, 44% yield): m.p. 259–261 °C, ^1^H NMR (300 MHz, CDCl_3_+pyridine-d_5_): δ 10.74 (s, NH), 8.88 (s, 1H), 7.63 (d, *J* = 9.03 Hz, 2H), 7.41 (s, 1H), 6.91 (d, *J* = 9.06 Hz, 2H), 6.82 (s, 1H), 5.34 (tt, *J* = 7.35, 1.47 Hz, 1H), 3.82 (s, 3H), 3.36 (d, *J* = 7.20 Hz, 2H), 1.80 (s, 3H), 1.72 (s, 1H) ppm., ^13^C NMR (75 MHz, CDCl_3_+pyridine-d_5_): δ 163.00 (C), 162.28 (C), 160.44 (C), 156.65 (C), 156.65 (C), 155.34 (C), 149.37 (CH), 134.40 (C), 131.01 (C), 130.16 (CH), 128.33 (C), 122.29 (2CH), 120.94 (CH), 113.20 (C), 111.74 (C), 101.71 (CH), 55.56 (CH_3_), 27.76 (CH_2_), 25.81 (CH_3_), 17.82 (CH_3_) ppm. IR: 3182.53 (O-H), 3111.40 (N-H), 3073.14 (C-H, aromatic), 2911.91 (C-H, aliphatic), 1695.83 (C=O, amide) cm^−1^; HREI-MS (*m*/*z*) calculated for C_22_H_22_O_5_N (M)^+^ 380.1493, found 380.1490.

#### 3.1.7. Synthesis of 8,8-Dimethyl-2-oxo-*N*-phenyl-2H,8H-pyrano[3,2-g]chromene-3-carboxamide **12**

A mixture of compound **14a** (1.7 g, 4.85 mmol) and DDQ (1.2 g, 5.28 mmol) in benzene (10 mL) was refluxed for 6 h, and the precipitate was filtered off. The filtrate was evaporated to dryness to afford the crude product, which was purified via column chromatography (silica gel, 10:1 hexane:EtOAc) to obtain a white solid of 8,8-dimethyl-2-oxo-N-phenyl-2H,8H-pyrano[3,2-g]chromene-3-carboxamide **12** (1.11 g, 66% yield): m.p. 187–188 °C, ^1^H NMR (300 MHz, CDCl_3_): δ 10.81(s, 1H), 8.89 (s, 1H), 7.75 (d, *J* = 1.14 Hz, 2H), 7.39 (t, *J* = 6.54 Hz, 2H), 7.30 (s, 1H), 7.18 (t, *J* = 1.14 Hz, 1H), 6.80 (s, 1H), 6.40 (d, *J* = 9.90 Hz, 1H), 5.78 (d, *J* = 9.96 Hz, 1H), 1.55 (s, 6H) ppm.

#### 3.1.8. Synthesis of 6-(3-Methyl-2-buteny1)-7-hydroxycoumarin **15**

Compound **13** (0.20 g, 0.73 mmol) in 2 mL quinoline containing 0.3 g Cu powder was heated for 3 min at 215–220 °C in an oil bath. The mixture was cooled to room temperature and diluted with CH_2_Cl_2_ (30 mL) prior to extraction with 10% HCI (2 × 30 mL) and then with water (30 mL). The solvent was evaporated, leaving a tacky orange solid, which was purified via column chromatography (silica gel, 1:1 hexane:EtOAc) to yield a cream solid of 6-(3-Methyl-2-buteny1)-7-hydroxycoumarin **15** (0.09 g, 53% yield), m.p. 134–136 °C (lit. 133 °C [24]), ^1^H NMR (300 MHz, CDCl_3_): δ 7.67 (d, *J* = 9.42 Hz, 1H), 7.20 (s, 1H), 7.06 (s, 1H), 6.24 (d, *J* = 9.42 Hz, 1H), 5.33 (tt, *J* = 8.73, 1.41 Hz, 1H), 3.38 (d, *J* = 7.23 Hz, 2H), 1.78 (s, 3H), 1.75 (s, 3H).

### 3.2. Determination of Antibacterial Activity

The antibacterial activities of coumarin derivatives **10**, **12**, **13**, **14a–g**, and **15** were evaluated against five reference standard bacteria, both gram-positive and gram-negative: *B. cereus* TISTR 2372, *B. subtilis* TISTR 001, *S. aureus* TISTR 2392, *E. coli* TISTR 073, and *S. typhimurium* TISTR 2519, using a standard microbroth dilution method [25]. 

The MIC values of coumarin derivatives **10**, **12**, **13**, **14a–g**, and **15** were determined through the microbroth dilution method in 96-well microtitre plates. The bacterial cultures were prepared from overnight cultures on nutrient broth (NB) at 37 °C for 24 h by diluting in NB compared with 0.5 McFarland. Coumarin derivatives **10**, **12**, **13**, **14a–g**, and **15** (5000 µg/mL) were prepared in EtOH, and 128 µg/mL of these were added to the first wells. Two-fold serial dilutions were prepared, and final concentrations of 128 to 1 µg/mL were achieved. The positive controls for penicillin G were determined, with the final concentrations from 128 to 1 µg/mL. In addition, an extra row of EtOH was used as a vehicle control to determine its possible inhibitory activity. Finally, 10 µL of bacterial suspension was added to each well. After the bacteria were incubated at 37 °C for 24 h, the microtitre plates were visually examined for bacterial growth; the growth rate was monitored at the optical density at 600 nm with a microplate reader. In each row, the well containing the lowest concentration that showed no visible growth was considered the MIC.

### 3.3. Cell Viability Assay

The cell viability assays of coumarin derivatives **10**, **12**, **13**, **14a**–**g**, and **15** were conducted against three cancer cell lines (HepG2, HeLa, and MDA-MB-231) and one normal cell line (LLC-MK2) using an MTT assay [26].

Stock solutions of coumarin derivatives **10**, **12a**, **13**, **14a**–**g**, and **15** were prepared in EtOH at a concentration of 5000 µg/mL. Prior to use, the stock solutions were further diluted to 128 µg/mL and added to the first wells. Two-fold serial dilutions were prepared, and final concentrations of 128 to 1 µg/mL in culture medium were achieved. Cells were seeded at a density of 5 × 104 cells/well in a 96-well plate and incubated for 16 h, followed by treatment with the test compounds. The control culture contained the carrier solvent of 2.5% dimethyl sulfoxide (DMSO). After 24 h, HepG2, HeLa, MDA-MB-231, and LLC-MK2 cells were then incubated with MTT (500 μg/mL) for 4 h. Then, DMSO was added to dissolve the blue formazan crystals formed, which were formed as a result of the action of cellular oxidoreductase enzymes on the MTT dye. Finally, the optical density at 450 nm was determined using a microplate reader.

## 4. Conclusions

We designed and synthesized a series of coumarin-3-carboxamides and evaluated their antibacterial and anticancer activities. The carboxylic acid at the C3 position of coumarins was necessary for the antibacterial activity, as seen for compounds 10 and 13, which showed moderate antibacterial activities against the tested gram-positive bacteria. Meanwhile, most of the tested compounds showed potent anticancer activity, and the 4-fluorophenyl coumarin-3′-carboxazine 4b was by far the most active anticancer, with activity comparable to that of the anticancer drug doxorubicin, and it had low cytotoxicity against a normal cell line. The molecular docking study revealed the binding to the active site of the CK2 enzyme, indicating that the presence of the phenyl carboxamide is important for anticancer activity.

## Data Availability

The data presented in this study are available on request from the corresponding author.

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
