# Peer review of "Synthesis and Biological Activity Evaluation of Coumarin-3-Carboxamide Derivatives"

_molecules, 2021, doi:10.3390/molecules26061653_

Round 1

Reviewer 1 Report

The originality of the paper lies in the synthesis of the amides which seems unknown. A synthetic effort has been made to prepare the different compounds and the experimental part is well described. In the naming of the compounds, the first letter encountered should be in capital. In reference 19, name has not to be in capital letters. Also a publication by Faulques M., ReneL., Royer R. , Averbeck D., Moradi M. ; European Journal of Medicinal Chemistry descibes thesynthesis of the acid. Synthesis and photo-​induced biological properties of demethyl derivatives of the natural pyranocoumarins, xanthyletin or seselin 

could be cited. Concerning the biological part, an study on antibacterial and anticancer activity has been done. One compound shows some interesting anticancer activity. Docking studies have been made on this point too.

Author Response

Dear Reviewer,

The English language and spelling have been carefully checked and corrected.

The naming of the compounds was used capital letter for the first letter, also with the reference 19.

Reference according to the previous synthesis of pyranocoumarins has been added.

Best Regards,

Waya Phutdhawong  

Reviewer 2 Report

This paper outlines the synthesis of some novel coumarin derivatives and their subsequent testing for antibacterial and anti-cancer activity. Molecuar docking experiment were then carried out to rationalize the anti-cancer activities.

Overall I thought this was a well written and executed paper that investigated the coumarin based compounds in a systematic way. The results point the way to some interesting compounds for development as well as a justification for the described results. I could not find any errors apart from a few minor spelling mistakes and would therefore recomend publishing this article as is (after a quick spell check/basic editorial corrections).

Author Response

Dear Reviewer,

I have been read through the manuscript carefully and I have edited some spelling mistake and also font/style of the text using track changes.

Best Regards,

Waya Phutdhawong

Reviewer 3 Report

The paper presents the kinetics of the drying process and the degradation of anthocyanins and bioactive properties of blackberry bagasse.

In my opinion the paper is worth studying and the manuscript contains enough original material. The experimental tests are carried out correctly using appropriate methods. The anticancer activity is quite interesting.

Minor corrections:

DOI numbers should be included in the references.

Text formatting should be carefully checked.

The language should be modified carefully.

Author Response

Dear Reviewer,

I have been read through the manuscript carefully and I have edited the format/style and also spelling mistake of the  text using track changes. The DOI of all references have been added.

Best Regards,

Waya Phutdhawong

Text formatting should be carefully checked.

The language should be modified carefully.